# Affinity3D: Propagating Instance-Level Semantic Affinity for Zero-Shot Point Cloud Semantic Segmentation

## ABSTRACT

Zero-shot point cloud semantic segmentation aims to recognize novel classes at the point level. Previous methods mainly transfer excellent zero-shot generalization capabilities from images to point clouds. However, directly transferring knowledge from image to point clouds faces two ambiguous problems. On the one hand, 2D models will generate wrong predictions when the image changes. On the other hand, directly mapping 3D points to 2D pixels by perspective projection fails to consider the visibility of 3D points in camera view. The wrong geometric alignment of 3D points and 2D pixels causes semantic ambiguity. To tackle these two problems, we propose a framework named Affinity3D that intends to empower 3D semantic segmentation models to perceive novel samples. Our framework aggregates instances in 3D and recognizes them in 2D, leveraging the excellent geometric separation in 3D and the zero-shot capabilities of 2D models. Affinity3D involves an affinity module that rectifies the wrong predictions by comparing them with similar instances and a visibility module preventing knowledge transfer from visible 2D pixels to invisible 3D points. Extensive experiments have been conducted on SemanticKITTI datasets. Our framework achieves state-of-the-art performance in two settings.

## CCS CONCEPTS

• **Computing methodologies** → **Object recognition**; **Image segmentation**; *Cluster analysis*; *Object identification*.

## KEYWORDS

Affinity, Zero-shot Semantic Segmentation, Point Cloud Semantic Segmentation, Pseudo Labels

## 1 INTRODUCTION

Autonomous driving and robotics are essential multimedia applications with multiple modality inputs like LiDAR and RGB images. 3D semantic segmentation, as an essential task in autonomous driving, provides point-level semantic information for planning and getting regions of interest. Previous methods mainly train models in a close-set manner, which pre-define classes. In existing outdoor datasets [3, 8], foreground instances are primarily categorized based on traffic participants, such as motor vehicles, non-motor vehicles, pedestrians, animals, etc. Objects like wheelchairs, cartons, etc., are not defined. When these objects appear on the road, perception

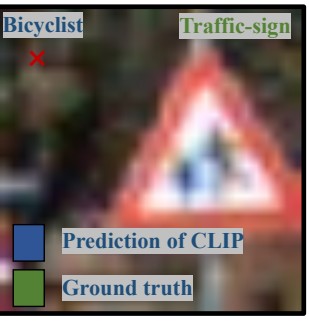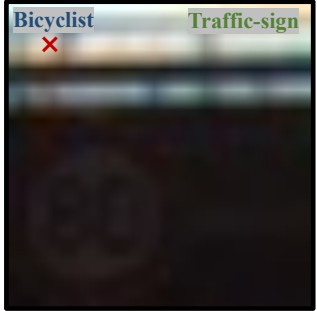

**Figure 1: The illustration of mistakes made by CLIP. The traffic signs in both two figures are wrongly classified as bicyclists.**

systems fail to correctly identify them, leading to potential safety issues disregarded by the decision-making system. Therefore, some methods [4, 14, 34, 46] have started to study endowing models with the ability to perceive unseen objects.

Unlike models trained in a close-set manner, zero-shot models can recognize novel classes without additional manual annotations. It is easier to train zero-shot models in 2D than in the 3D domain because 2D models can improve generalization performance from large-scale datasets [14, 16] for downstream tasks. Although a lot of synthetic data could be used for pretraining, significant differences exist between synthetic shapes and real-world scenes, and there are clear distinctions in scale between indoor and outdoor environments. So, it is hard to pre-train fundamental models with excellent zero-shot capability in the 3D domain.

Existing methods for recognizing unseen objects in 3D primarily involve transferring knowledge from language or images [5, 22, 27, 29]. Directly training a language-3D-aligned model without a large-scale dataset is challenging, so researchers [39, 47, 53] try to align outputs or features of 3D models with vision-language models like CLIP [34] to enhance generalization. However, these methods assume CLIP generated accurate predictions, which is unrealistic. Models like CLIP struggle to represent instance features effectively and may produce wrong predictions, especially with slight image variations or challenging conditions, as illustrated in Fig. 1. In the SemanticKITTI, there are about 3.11% similar instances wrongly classified by CLIP. The **wrong predictions** from CLIP can confuse the 3D network during output alignment, reducing the discriminative capability of the 3D network.

The other ambiguous problem of transferring vision-language models to 3D scenes is caused by the **wrong geometric alignment** between 3D points and pixels on 2D images. The wrong geometric alignment is due to the different visibility between LiDAR and the camera. Some objects may be visible in LiDAR but occluded in the image. As shown in Fig. 3, if 3D points of ground are misaligned with a vehicle, 3D models will transfer the vehicle semantics to

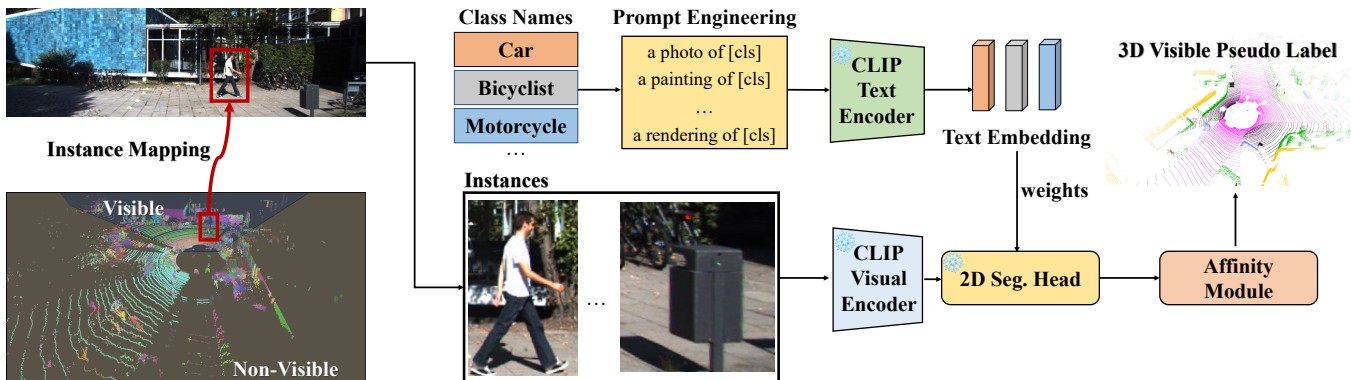

**Figure 2: The instance pseudo labels generation pipeline of Affinity3D. The 3D points are clustered into superpoints. Each superpoint is treated as an instance. The projection of superpoint crops the image of an instance. The cropped images are sent to the CLIP visual encoder, obtaining CLIP feature $F_{CLIP}$. The prompt engineering fills class names into templates to create various prompts. The CLIP text encoder encodes prompts. For each class, an average pooling is applied for their various features of prompts to get its text embedding. The text embeddings of all the classes are utilized as weights for segmentation heads to get initial pseudo labels. The affinity module refines the initial pseudo labels by comparing them with similar instances.**

ground points. It will cause the 3D model confuse of discriminating between vehicle and ground. According to our assessments, approximately 34% of the points in SemanticKITTI are wrongly matched. The wrong geometric alignment leads to inappropriate knowledge transferring between images and point clouds.

To deal with the wrong predictions, we propose an affinity module to rectify mistakes by incorporating predictions of similar objects. In 2D scenes, affinity is a common technique for refining pseudo labels. However, due to the lack of reliable similarity measurement methods in point clouds, affinity is rarely utilized in 3D methods. To achieve reliable affinity propagation, locating object instances is essential. Therefore, we propose an instance generation module to separate objects in point clouds. Specifically, we cluster the point cloud into superpoints and treat the superpoints as instances, as point clouds possess excellent geometric separation capabilities. To construct reliable affinity, we extract the features of instances from 2D image patches corresponding to superpoints. In such a way, the strong zero-shot learning abilities of vision-language models [6, 14, 18, 34] can be well utilized. Our instance generation module enables affinity propagation in 3D scenes, facilitating better knowledge transfer from images to point clouds.

To address the wrong geometric alignment issue, we propose a visibility measurement module to estimate the visibility of points from the camera perspective. The wrong geometric alignment typically occurs when the points belonging to invisible objects are wrongly projected onto a visible object. The depths of wrongly projected points are usually larger than the depth of points from visible objects. Therefore, our visibility measurement module first estimates the depth $D_{sp}$ of the visible object and treats the points with depths exceeding $D_{sp}$ too much as invisible. After measuring the visibility, we only distill the knowledge of 2D images to visible 3D points for reliable knowledge transfer.

Incorporating the aforementioned modules, we propose a transductive zero-shot point cloud semantic segmentation framework [5,

27] named Affinity3D. We evaluated Affinity3D on SemanticKITTI, and extensive experiments show that our Affinity3D outperforms existing state-of-the-art methods without increasing inference time.

Our contributions can be summarized as follows:

(1) An affinity module is proposed to refine wrong predictions of the visual-language model and generate accurate pseudo labels.

(2) A visibility measurement module is proposed to avoid misaligned non-visible 3D points to 2D pixels and achieve reliable knowledge transfer.

(3) We propose Affinity3D, which demonstrates state-of-the-art results on the SemanticKITTI dataset. Notably, it achieved 62.78% mIoU under a generalized zero-shot learning setting.

## 2 RELATED WORK

### 2.1 Close-set 3D semantic segmentation

The objective of 3D semantic segmentation is to classify each point in a scene. Earlier methods build models based on Point-Net [32], such as PointNet++ [33], DGCNN [31], RS-CNN [25], PointASNL [43], PointConv [40], KPConv [38], and PointTransformer [48]. However, those methods may perform better in 3D object classification or 3D indoor semantic segmentation but get poor results on outdoor scenes. Outdoor scenes are larger and more sparse than synthetic shapes or indoor scenes. To decrease runtime latency, researchers propose projection-based methods. Those methods project 3D points into a 2D view, like range-view [15] and polar-view, and then employ 2D segmentation structures. However, due to the lack of 3D operation, projection-based methods make it hard to model 3D relations, which leads to poor segmentation performance. Some methods investigate how to build a better representation in voxels or from multi-representation [21, 24, 44, 44, 52].

Although closed-set 3D semantic segmentation has made significant progress in network architecture and modal fusion, it cannot recognize unseen objects.

## 2.2 Zero-shot 3D semantic segmentation

Zero-shot 3D semantic segmentation methods can be categorized into generative, projection-based, and transfer-based approaches. 3DGenZ [29] and SeCondPoint [22] employ conditional generative networks, using class names as conditions to generate features for unseen samples, which are then provided to the classifier for training. Generative models trained solely on names and seen class samples may struggle to produce high-quality unseen class results during generation, leading to low zero-shot capability. Projection-based methods [5, 27] learn the mapping between 3D and text on seen classes, but this mapping requires substantial support from seen classes to generalize to unseen ones. Transfer-based methods transfer knowledge from 2D vision-language models with excellent zero-shot capabilities to 3D. Wang [39] transfers knowledge from CLIP [34] to 3D using contrastive learning at different scales and generates corresponding pseudo labels by MaskCLIP [51] to supervise the training of 3D models. CLIP2Scene [4] proposes semantic and spatial-temporal consistency regularization to pre-train the 3D network. OpenScene co-embeds dense 3D point features with image pixels and text in the CLIP feature space, enabling zero-shot training and open-vocabulary queries.

As a promising semi-supervised learning method, Pseudo-labeling techniques have been widely utilized in 2D zero-shot learning for self-training. The core idea of this technique is to employ a pre-trained model on unlabeled data to generate predicted labels, which are then regarded as true labels (i.e., "pseudo labels"). These pseudo labels are integrated into subsequent training processes, leveraging many unlabeled data for model optimization. In order to obtain pseudo labels, RegionCLIP [50] builds a concept pool using frequently occurring nouns in image captions. It extracts pseudo labels by performing classification using the concept pool and CLIP. GOOD [10] employs a pre-trained deep model to provide additional depth modalities as pseudo labels. XPM [12] proposes a cross-modal pseudo-labeling framework, which generates training pseudo masks by aligning word semantics in captions with visual features of object masks in images. Zhao et al. [49] proposed a split-and-fusion (SAF) head designed to remove the noise in the localization of pseudo labels. Besides, the training of visual-language-models [14, 19, 23] also benefits from utilizing a data augmentation engine, in which pseudo-label generation techniques are similarly employed.

Compared with CLIP2Scene [4] and OpenScene [30], our Affinity3D does not require maintaining complex spatial-temporal consistency relationships at training and does not need heavier open-vocabulary image segmentation models during inference time. Besides, previous methods mainly assume that vision-language models will not generate incorrect knowledge. Vision-language models will also make mistakes and transfer erroneous knowledge to 3D models. Our Affinity3D refines responses generated from vision-language models by comparing them with similar instances.

## 2.3 Cross-modal knowledge transfer

Transferring knowledge between different modalities is effective in 3D pretraining for improving downstream tasks. Methods of feature-level alignment mainly construct consistency loss on intermediate representation. Some researchers apply contrastive or distillation loss between 2D images and 3D point clouds [13, 26, 28, 35, 42, 45, 46]. Some methods like Image2point [41] assume that knowledge is stored in parameters, so they transform 2D knowledge into 3D by inflating of 2D parameters to 3D sparse convolution. Others try to align the image language and 3D features into a uni-representation [11, 17].

However, those methods need more exploration of transferring pathways and considering visibility change problems when mapping 3D points with 2D pixels. The transferring path without careful consideration will introduce erroneous semantics into the 3D model, thereby diminishing zero-shot capabilities. Our Affinity3D proposes a visibility measurement module that facilitates the visibility change problem.

## 3 METHOD

Our proposed Affinity3D aims to transfer knowledge from CLIP [34] to 3D, utilizing affinity to refine the wrongly generated predictions from CLIP and visibility to eliminate semantic ambiguity between visible 3D points and non-visible ones. Affinity3D contains an instance generation module, a pseudo labels generation module, a visibility measurement module, an affinity module, and a knowledge transfer module. In Fig. 2, the instance generation module clusters 3D points as superpoints. The 3D superpoints are treated as instances and sent to the pseudo labels generation module to obtain initial pseudo labels. The affinity module refines the pseudo labels by comparing them with similar objects. The knowledge transfer module distills knowledge from 2D pixels to visible 3D points determined by the visibility module and supervises the network by the refined pseudo labels.

## 3.1 Preliminary

The generalized zero-shot 3D semantic segmentation split all classes $C$ into seen classes $S$ and unseen classes $U$, where $S \cap U = \emptyset$ and $S \cup U = C$. The seen classes have labels, while the unseen annotations are not provided. In the training phase, point clouds $P \in \mathbb{R}^{N \times 4}$, 2D images $I \in \mathbb{R}^{H \times W \times 3}$, and class names are required as input. Only point clouds $P$, class names, and the 3D model are available at inference time. The 3D model needs to segment both seen and unseen classes correctly.

As shown in Fig. 3, the point cloud $P$ are sent to 3D Backbone and encoded as $F_{3D}^l \in \mathbb{R}^{N \times C_{3D}^l}$ at $l$-th layer. The image $I$ are encoded by 2D backbone into feature map $F_{2D}^l \in \mathbb{R}^{H^l \times W^l \times C_{2D}^l}$ at $l$-th layer. For a given point $p_i$, its 3D feature at $l$-th layer is $F_{3D}^{i,l}$ and the corresponding 2D feature at $l$-th layer is $F_{2D}^{i,l}$. The corresponding 2D feature for points $p_i$ is formulated as follows:

$$[u, v, 1]^T = \frac{1}{d} \cdot K \cdot E \cdot [x, y, z, 1]^T, F_{2D}^{i,l} = F_{2D}^l[u, v], \qquad (1)$$

where $[x, y, z]$ is the position of point $p_i$, $d$ is the depth of point $p_i$, $[u, v]$ is the pixel position of point $p_i$, $K$ is the intrinsic matrix, and $E$ is the extrinsic matrix of the camera.

The prompt engineering integrates and expands class names based on different templates, such as "a photo of [cls]," where [cls] is the inserted class name. After multiple prompts are generated from various templates for the same class, the class embedding is obtained by averaging the text features from the CLIP text encoder.

It can be formulated as follows:

$$emb^{c_i} = \frac{1}{N_{c_i}} \sum_{j=1}^{N_{c_i}} \text{CLIP}_{\text{text}}(template_j^{c_i}(C^i)), \tag{2}$$

$$EMB^{\Omega} = \{emb^{c_i}|c_i \in \Omega\}, \Omega \in \{C, S, U\}, \tag{3}$$

where $template_j^{c_i}(\cdot)$ is the $j$-th text editing function for class $c_i$, $C^i$ is the name of class $c_i$, $\text{CLIP}_{\text{text}}$ is the CLIP text encoder, and $N_{c_i}$ is the number of template for class $c_i$. $EMB^C$ represents the text embeddings for all classes, and $EMB^S/EMB^U$ denotes the text embeddings for seen/unseen classes.

## 3.2 Instance generation module

The geometric spatial structure inherent in point clouds allows for superior instance separation through clustering compared to 2D images. Therefore, our instance generation module clusters point clouds into superpoints. Each superpoint is regarded as an individual instance. After projection, the superpoints can be enclosed by a 2D bounding box. For bounding boxes smaller than $30 \times 30$, the height and width are set to $30 \times 30$. Superpoints with less than 2 points are regarded as invalid. As shown in Fig. 2, the patch corresponding to the 2D bounding box of each superpoint $SP_j$ is encoded through CLIP visual encoder to obtain the feature $F_{CLIP}^{SP_j}$. Consequently, each instance has a superpoint, cropped 2D patch, and CLIP feature.

## 3.3 Pseudo label generation module

After the instance generation module, we ignore the instances corresponding to seen classes and keep the remaining instances that fall inside images. Let $F_{CLIP} \in \mathbb{R}^{N_I \times C_I}$ be the matrix consisted of all feature $F_{CLIP}^{SP_j}$ for kept superpoints. $N_I$ is the number of kept instances, $C_I$ is the channel of feature. As shown in Fig. 2, the pseudo labels of kept instances are computed based on the response of CLIP features to text embedding. It can be formulated as:

$$Y_{kept} = softmax(F_{CLIP} \cdot EMB^U), \tag{4}$$

where $Y_{kept}$ is the semantic prediction for superpoints.

## 3.4 Affinity module

The initial prediction $Y_{kept}$ is noisy due to image jitter and occlusion. In the transductive zero-shot setting, unseen classes are not provided labels during training. Therefore, we can enhance the quality of initial prediction to mitigate noise by deeply exploring the relationships among samples. The similarity between instances can rectify this noise. As illustrated in Fig. 4, if a particular instance yields a wrong prediction at time $t$, corrections can be made by comparing instances across preceding and subsequent frames or considering contextual instances. The similarity between instances is defined as affinity, which can be formulated as:

$$A = scale \cdot F_{CLIP}^{SP_i} \cdot F_{CLIP}^{SP_j}, \tag{5}$$

where $F_{CLIP}^{SP_i}/F_{CLIP}^{SP_j}$ is the CLIP feature of instance $i/j$, and $scale$ is the amplify factor.

The affinity matrix $W$ between multiple instances is as follows:

$$W = softmax(scale \cdot F_{CLIP}F_{CLIP}^T), \tag{6}$$

where softmax is used for normalization. The propagation of instance affinity is conducted in the form of random walks. It can be formulated as:

$$T = D^{-1}W^{\circ\beta}, where\ D_{ii} = \sum_j W_{ij}^{\circ\beta}. \tag{7}$$

In Eqn. (7), $W^{\circ\beta}$ represents the element-wise power of $W$ to the exponent of $\beta$, where $D$ is the normalized diagonal matrix. The diagonal values in $D$ are the sums of the corresponding rows in $W^{\circ\beta}$. $T$ is the propagation matrix. The instance pseudo labels can be updated as follows:

$$Y_{kept}' = T \cdot Y_{kept}, \tag{8}$$

where $Y_{kept}'$ is the update response matrix. The refined pseudo labels $PL'$ provides a semantic class for each point. The refined pseudo labels for seen labels are filled with ground truths. Points belonging to the kept superpoints are filled with $Y_{kept}'$. Others are set to ignoring labels.

For a given instance, instances in adjacent frames contribute more to affinity propagation. Therefore, we adopt a queue-based approach, where once the number of instances in the queue reaches a pre-defined number, all instances are popped from the queue. Then, the pseudo labels of instances are refined according to Eqn. (8).

## 3.5 Visibility measurement module

Although the noises in pseudo labels are reduced after passing through the affinity module, 3D points in outdoor scenes are still affected by changes in visibility. The projection mapping between point clouds and images often adopts the perspective projection method (Eqn. (1)). However, the visibility of objects in LiDAR and cameras differs. Some objects may be visible in point clouds but occluded in images. It can be seen in Fig. 3 that the differences in visibility between LiDAR and the camera can result in wrong mappings between 3D points and 2D pixels, where 3D points belonging to the road may map to a car in the image. Wrong mappings lead to incorrect pseudo labels or inaccurate 2D-to-3D knowledge transfer. We measure the visibility of 3D points in the camera view to address the misaligned issue. Specifically, the calculation method involves first obtaining superpixel segmentation results from 2D images. Then, mapping relationships between superpixels and 3D points are computed by perspective projection (Eqn. (1)). For each superpixel, its depth $D_{sp}$ is represented as the minimum depth of the points set that belong to that superpixel. The depth $D_{S_p}$ of superpixel is formulated as:

$$D_{S_p} = \text{scatter}_{\min}(D, S_p), \tag{9}$$

where $D$ is the depth vector of 3D points, $S_p$ is the superpixel identification (ID) vector of 3D points. The element of the superpixel ID vector is computed based on the direct perspective projection mapping between 3D points and superpixel. If a 3D point falls outside the image bound, its superpixel ID is $-1$. '$scatter_{min}$' refers to the scatter minimum function[1], which calculates the minimum value of the same identification.

---

[1] https://pytorch-scatter.readthedocs.io/en/latest/functions/scatter.html

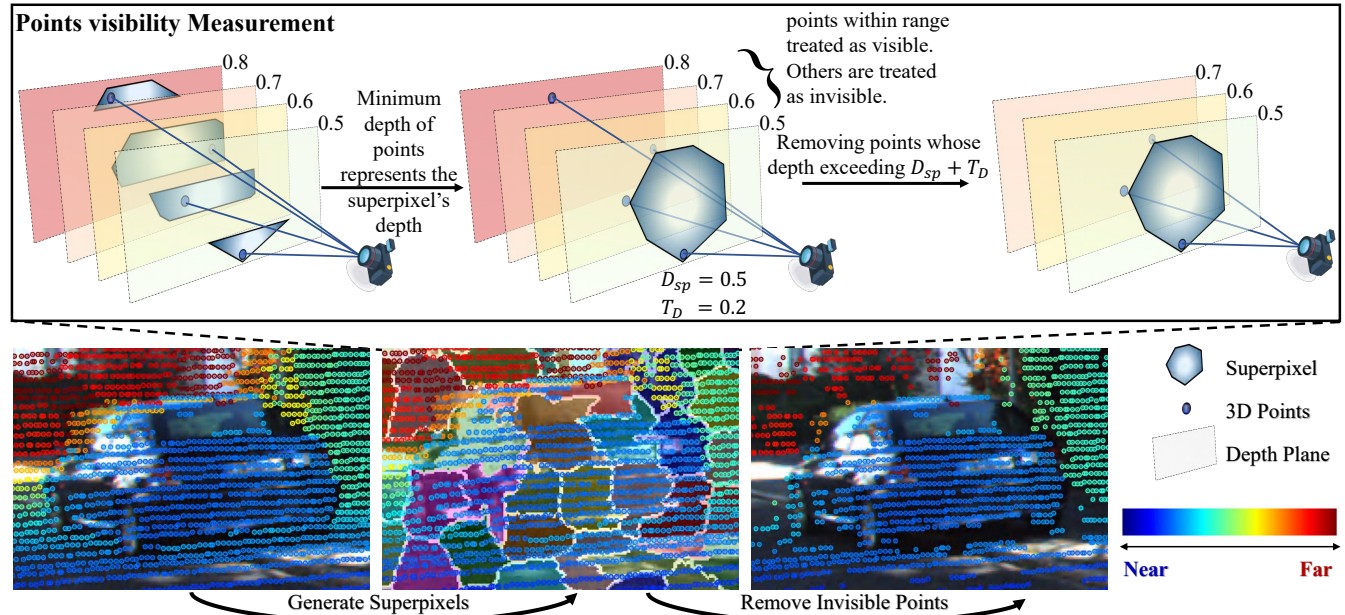

**Figure 3: The visibility measurement module of Affinity3D. 3D Points are projected to 2D pixel by perspective projection (Eqn. (1)). The 3D points that fall outside the image bound are considered invisible. The superpixel segmentation algorithm segments images into several superpixels. For each superpixel, the minimum depth of 3D points that belong to it represents the superpixel's depth $D_{sp}$. Points with a depth exceeding $D_{sp} + T_D$ are treated as invisible. The invisible 3D points are ignored in pseudo-label generation and knowledge-transferring modules.**

Usually, local similar pixels are adjacent in 3D space. So, if the depth $D$ of points greater than $D_{sp} + T_D$, it will be determined as invisible. $T_D$ is a predefined threshold. The visibility $Vis$ is formulated as follows:

$$Vis = \mathbb{1}(D - D_{S_p}[S_p] < T_D \ \& \ S_p > -1), \quad (10)$$

where $\mathbb{1}(\cdot)$ denotes the visibility function.

## 3.6 Knowledge transfer module

Due to the lack of large-scale pre-training datasets in point clouds, many methods incorporate data augmentation during training to enhance the generalization of 3D models. In order to improve the network's generalization ability and discriminative capacity towards novel classes, Affinity3D incorporates the Multi-scale Fusion-to-Single Knowledge Distillation (MSFSKD) in 2DPass [42] during training, gradually transferring knowledge from images to point clouds. However, MSFSKD still needs to select knowledge transfer pathways. To avoid incorrect semantic correspondences, we utilize Eqn. (10) to filter out invisible points and only construct knowledge transfer loss for visible points.

Specifically, the feature $F_{3D}^{i,l}$ of a visible 3D point $p_i$ at $l$-th layer is transformed to 2D space through a 2D learner that is built by multi-layer perceptron (MLP), resulting in $F_{2Dlearner}^{i,l}$. The 2D feature $F_{2D}^{i,l}$ of a visible 3D point $p_i$ at $l$-th layer and its corresponding $F_{2Dlearner}^{i,l}$ are concatenated into $F_{2D3D}^{i,l}$. In order to further mix 2D and 3D features of visible 3D points $p_i$, each channel in the $F_{2D3D}^{i,l}$ is selected through a gated module. Specifically, its transformation formula is as follows:

$$F_{gated}^{i,l} = \text{sigmoid}\left(\text{MLP}_G\left(\text{MLP}_{2D3D}\left(F_{2D3D}^{i,l}\right)\right)\right) \quad (11)$$

$$\dot{F}_{2D3D}^{i,l} = \text{ReLU}\left(F_{gated}^{i,l} \odot \text{MLP}_{2D3D}\left(F_{2D3D}^{i,l}\right)\right) + F_{2D}^{i,l}, \quad (12)$$

where $\text{MLP}(\cdot)$ is the multi-layer perceptron. sigmoid is the activation function. $\dot{F}_{2D3D}^{i,l}$ is the final 2D-3D fusion feature. The segmentation head is used to obtain the 3D prediction vector $Y_{3D}^{i,l}$ and 2D-3D fusion prediction vector $Y_{2D3D}^{i,l}$ of $p_i$ at $l$-th layer. It has to be noted that the 2D and 3D segmentation heads are composed of text embeddings as weights. The final prediction can be formulated as follows:

$$Y_{3D}^{i,l} = \text{softmax}\left(F_{3D}^{i,l} \cdot EMB^C\right), \quad (13)$$

$$Y_{2D3D}^{i,l} = \text{softmax}\left(\dot{F}_{2D3D}^{i,l} \cdot EMB^C\right). \quad (14)$$

A Kullback–Leibler (KL) divergence loss is adopted between the 2D-3D fusion prediction and the 3D prediction. It can be formulated as follows:

$$L_{learner} = \frac{1}{N_v} \sum_{p_i \in V_p} KL(Y_{3D}^{i,l}, Y_{2D3D}^{i,l}), \quad (15)$$

$$V_p = \{p_i | Vis[i] = 1\}, \quad (16)$$

where $N_v$ is the number of visible 3D points, $V_p$ is the set containing all the visible 3D points.

## 3.7 Joint training

3D point clouds are susceptible to class imbalance issues, where the network tends to overfit categories with dominant points. Lovasz

loss [2] is a class-balanced loss in a mini-batch, which achieved mean intersection-over-union loss for multi-class semantic segmentation. The weighted cross entropy loss utilizes the point frequency statistics on the dataset as weights to balance the contribution of different classes. Therefore, Lovasz and weighted cross entropy loss supervise the classification prediction vectors. The final loss function formula is:

$$F_{3D} = \text{MLP}_{Seg_{3D}} \left( \text{concat} \left( F_{3D}^0, ..., F_{3D}^{N_L} \right), P_{Seg_{3D}} \right) \quad (17)$$

$$Y_{3D} = \text{softmax} \left( F_{3D} \cdot EMB^C \right), \quad (18)$$

$$L_{3D} = \alpha \, \text{lovasz} \left( Y_{3D}, PL' \right) + CE \left( Y_{3D}, PL', w \right) \quad (19)$$

$$L = L_{3D} + \gamma L_{learner} \quad (20)$$

where $MLP_{Seg_{3D}}$ is the multi-layer perceptron with parameters $P_{Seg_{3D}}$, used to reduce the dimension of concatenated features from multiple layers. $N_L$ is the number of layers, $\alpha$ and $\gamma$ are the balance factor, and $w$ is the class weight of weighted cross entropy.

## 4 EXPERIMENTS

### 4.1 Dataset and metric

Our experiments were conducted on the semanticKITTI [1] dataset. The semanticKITTI dataset consists of 22 sequences of point clouds, where sequences 00 to 10 were used for training, sequence 08 was used for validation, and sequences 11 to 21 were used for double-blind test evaluation. We mainly reported the metric of mean intersection over union (mIoU), defined as the average IoU over all classes. Two experimental settings were conducted: generalized zero-shot and annotation-free. In the generalized zero-shot setting, we divided all classes of SemanticKITTI into seen and unseen ones, where motorcycle, truck, bicyclist, and traffic signs were treated as unseen classes. The average IoU over seen and unseen classes were also reported. Harmonic mean IoU (hIoU) was also utilized to evaluate the general zero-shot performance of models. Harmonic mIoU considered both mIoU for seen classes and unseen classes, which was formulated as:

$$hIoU = \frac{2 \times mIoU_{seen} \times mIoU_{unseen}}{mIoU_{seen} + mIoU_{unseen}}, \quad (21)$$

where $mIoU_{seen}/mIoU_{unseen}$ represented the mIoU for seen/unseen classes. Following the setting of TCKZ [39], we only evaluated the front view of LiDAR points on SemanticKITTI.

### 4.2 Implementation details

To save memory, we applied random cropping to the images. During training, we involved data augmentation on the point clouds to prevent overfitting on the training set. The data augmentation included global random translation, scaling, and downsampling. The global random translation range was [-0.5, 0.5], the global random scaling range was [0.95, 1.05], and the 70% global random downsampling was performed for point clouds. Regarding the 2D image, we have implemented global random flipping and color jitter. The $\beta$ in the random walk was 2, and the $T_D$ in Eqn. (10) was 0.5. For a fair comparison with the previous method, the backbone of the 3D model was SPVCNN [37]. The CLIP model was ViT-B/32 in our pseudo label generation module. In the knowledge transfer module, the 2D image encoder was ResNet-34 [9]. When annotations were available

**Table 1: Comparisons with other state-of-the-art methods on SemanticKITTI validation set under a generalized zero-shot setting. 'FS' means fully supervised, and 'ZS' represents zero-shot. 'Ann.' is the abbreviation of annotation. 'S'/'U' means providing seen/unseen class labels during training.**

| Setting | Ann. | | Method | mIoU | | | hIoU |
|---|---|---|---|---|---|---|---|
| | S | U | | Seen | Unseen | All | |
| FS | ✓ | ✓ | SPVCNN [37] | 66.97 | 60.35 | 65.58 | 63.49 |
| | ✓ | | SPVCNN [37] | 62.31 | 0 | 49.19 | 0 |
| ZS | ✓ | | MaskCLIP-3D+ [51] | 45.88 | 19.58 | 40.34 | 27.44 |
| | ✓ | | 3DGenZ [29] | 41.40 | 10.80 | 35.00 | 17.10 |
| | ✓ | | TGP [5] | 54.60 | 17.30 | 46.70 | 26.30 |
| | ✓ | | TCKZ [39] | 61.31 | 46.50 | 58.19 | 52.89 |
| | ✓ | | Affinity3D | **61.41** | **66.06** | **62.78** | **63.65** |

**Table 2: Comparisons with other state-of-the-art methods on SemanticKITTI validation set under an annotation-free setting. TTA means test-time augmentation.**

| Method | Input | SemanticKITTI |
|---|---|---|
| MaskCLIP-3D+ [51] | Camera+LiDAR | 8.99 |
| TCKZ [39] | LiDAR | 13.17 |
| Affinity3D | LiDAR | 18.48 |
| Affinity3D+TTA | LiDAR | 19.40 |

for seen classes in the instance generation module, DBSCAN [7] clustered unseen classes. When no annotations were available, the boundary preserved supervoxel segmentation (BPSS) [20] method was employed to generate superpoints. The $\alpha$ in Eqn. (19) was 2.0 and the $\gamma$ in Eqn. (20) was 0.05.

### 4.3 Comparisons with the state-of-the-art methods

Table 1 demonstrated the comparison with the prior state-of-the-art methods on the SemanticKITTI validation set under a generalized zero-shot setting. Our method achieved the best mIoU results without introducing more inference time. Compared with previous methods, Affinity3D achieved a 19.56% increase in mIoU for unseen classes and an overall improvement of 4.59% across all classes. Additionally, considering metrics for seen and unseen classes, the hIoU improved by 11.44%. Besides, the mIoU across all the classes of Affinity3D approached the level of full supervision.

The results of our Affinity3D and previous methods under an annotation-free setting were shown in Table 2. It was observed that the performance of Affinity3D improved compared with that of TCKZ [39]. If the distribution of point clouds at test time differed from training, it could lead to performance degradation. Therefore, we included test-time data augmentation (TTA) in Table 2 to address this issue. Specifically, TTA included randomly rotating the point clouds and then averaging the results predicted by the model to obtain the augmented prediction result. Affinity3D with TTA achieved a 0.92% absolute improvement in Table 2.

### 4.4 Ablation study

Since all classes were unseen under the annotation-free setting, the mIoU for seen classes and hIoU were not reported, and the mIoU

**Table 3: The ablation study on SemanticKITTI validation set under a generalized zero-shot setting. 'MC' means that pseudo labels are generated by MASKCLIP. 'CI' represents the pseudo labels generated by our instance generation module and pseudo label generation module. KT indicates whether a knowledge transfer module was involved during training. 'GZS'/'AF' represents generalized zero-shot/annotation-free. 'Aff.'/'Vis.' represents the abbreviations of affinity/visibility.**

| Setting | Labels | Aff. | Vis. | KT | Seen | Unseen | All | hIoU |
|---------|--------|------|------|-----|------|--------|-----|------|
| GZS | MC | | | | 55.88 | 35.41 | 52.13 | 43.35 |
| | MC | | ✓ | | 56.45 | 35.13 | 52.50 | 43.31 |
| | MC | | | ✓ | 57.95 | 36.62 | 54.05 | 44.88 |
| | MC | | ✓ | ✓ | 57.89 | 40.31 | 54.76 | 47.53 |
| | CI | | ✓ | | 58.73 | 62.83 | 59.87 | 60.71 |
| | CI | ✓ | ✓ | | 59.00 | 65.71 | 60.76 | 62.17 |
| | CI | | ✓ | ✓ | 60.88 | 67.59 | 62.66 | 64.06 |
| | CI | ✓ | ✓ | ✓ | 61.41 | 66.06 | 62.78 | 63.65 |
| AF | MC | | | | - | 11.37 | 11.37 | - |
| | MC | | ✓ | | - | 11.65 | 11.65 | - |
| | MC | | | ✓ | - | 12.96 | 12.96 | - |
| | MC | | ✓ | ✓ | - | 13.78 | 13.78 | - |
| | CI | | ✓ | | - | 16.60 | 16.60 | - |
| | CI | ✓ | ✓ | | - | 17.41 | 17.41 | - |
| | CI | | ✓ | ✓ | - | 18.06 | 18.06 | - |
| | CI | ✓ | ✓ | ✓ | - | 18.48 | 18.48 | - |

for unseen and all classes were the same. To ensure fairness, the backbone used for the 3D network was SPVCNN [37], and the image branch added for knowledge transfer exists only during training. We demonstrated the ablation study under generalized zero-shot and annotation-free settings in Table 3.

*Effect of instance generation module.* In Table 3, we conducted an ablation experiment on pseudo labels in rows 4, 7 and 12, 15. Compared with MaskCLIP, our instance generation module and pseudo label generation module could generate instance-level pseudo labels that improved the final performance. Specifically, introducing instances (row 4 vs row 7) led to an absolute improvement of 27.28% in mIoU for unseen classes under the generalized zero-shot setting. Under the annotation-free setting, there was an absolute mIoU improvement of 4.28% (row 12 vs row 15). CLIP was a model trained with image-level supervision, lacking precise object localization capabilities. Therefore, our instance generation module utilized the excellent object separation properties of 3D to aggregate objects and generate pseudo labels. It greatly enhanced CLIP's perceptual capabilities for objects and enabled more accurate knowledge transfer.

*Effect of affinity module.* To validate the effectiveness of our affinity module, we conducted ablation experiments with four different settings. Incorporating our affinity module resulted in approximately 1% (row 5 vs row 6 and row 13 vs row 14) absolute improvements in mIoU for both generalized zero-shot and annotation-free settings. Compared with the full model, when the affinity module was removed, the mIoU decreased by 0.12%/0.42% (row 7 vs row 8/row 15 vs row 16) for all classes under the generalized zero-shot/annotation-free setting. It seems that when all three modules were employed simultaneously, the efficacy of our affinity module

**Table 4: Pseudo label quality on SemanticKITTI training set. 'CI' represents the pseudo labels generated by our instance generation and pseudo label generation module. 'GZS'/'AF' represents generalized zero-shot/annotation-free.**

| Method | Affinity | Setting | Accuracy |
|--------|----------|---------|----------|
| CI | | GZS | 12569/15043=83.55% |
| CI | ✓ | GZS | 13084/15043=86.98% |
| CI | | AF | 384543/1107790=34.71% |
| CI | ✓ | AF | 418973/1107790=37.82% |

tended to diminish. The reason may be that the affinity, visibility, and knowledge transfer modules serve similar purposes, aiming to eliminate semantic ambiguity from different perspectives. Therefore, the effects of individual modules may not be remarkable when working together.

*Effect of visibility measurement module.* Our visibility measurement module could serve to judge whether a pseudo-label should be kept for self-training (Eqn. (15)) and knowledge transfer (Eqn. (19)). The invisible points are set to the ignoring label for self-training. The knowledge transfer was only applied to visible points determined by the visibility measurement module. As shown in Table 3, compared with MaskCLIP, our visibility module resulted in an absolute improvement of 0.37% (row 1 vs row 2) in mIoU for all classes in a generalized zero-shot setting. In the annotation-free setting, there was an absolute improvement of 0.28% (row 9 vs row 10). Besides, when our visibility module was served to guide the knowledge transfer, an absolute improvement of 2.65% (row 3 vs row 4) for hIoU was achieved in a generalized zero-shot setting. For the annotation-free setting, there was an absolute mIoU improvement of 0.82% (row 11 vs row 12). The above results verified our visibility module's effectiveness as a guide for noise filtering.

## 4.5 Pseudo labels quality

To better understand the effectiveness of our affinity module on the quality of the pseudo labels, we conducted additional experiments and reported the results in Table 4. We adopt CLIPInstance as the baseline model, which involved passing the 2D image patches of individual instances through CLIP ViT-B/32 to extract image features and computing the maximum response with text embedding to obtain predictions. Pseudo labels for points were assigned based on the predictions of the instances to which they belong. We applied affinity by Eqns. (7) and (8) to CLIPInstance in both the generalized zero-shot and the annotation-free settings. Compared with CLIPInstance, our affinity module led to an approximately 3% absolute accuracy improvement of pseudo labels under both settings. Pseudo labels with less noise could reduce the introduction of wrong semantics during training, thereby improving segmentation performance.

## 4.6 Visualization results

We demonstrated some instances' maximum affinity target in Fig. 4. It can be seen that a greater presence of semantically similar parts in images corresponded to higher affinity. Conversely, larger semantic differences between images resulted in lower affinity. It indicated

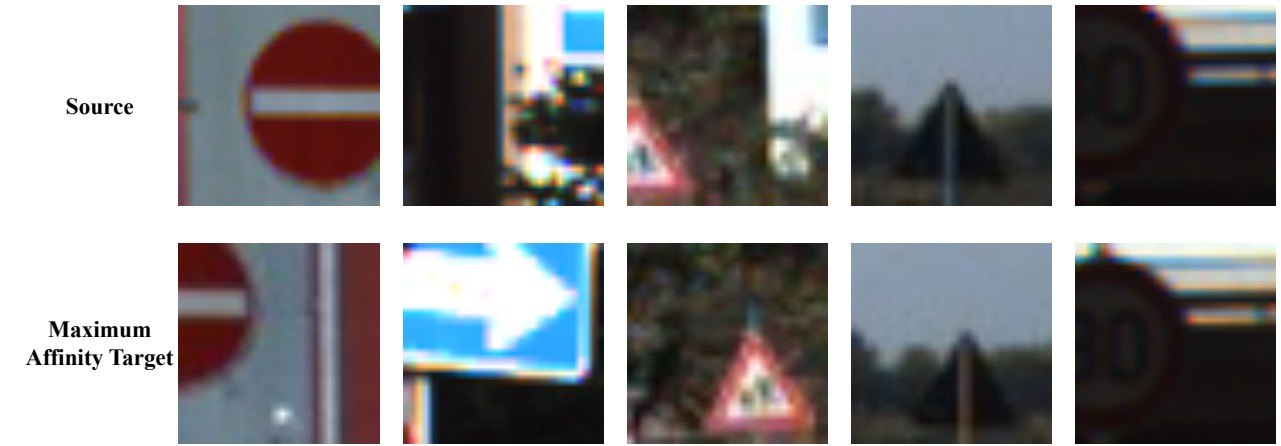

**Figure 4: The visualization results of affinity. The source images (row 1) and their maximum targets (row 2) are presented.**

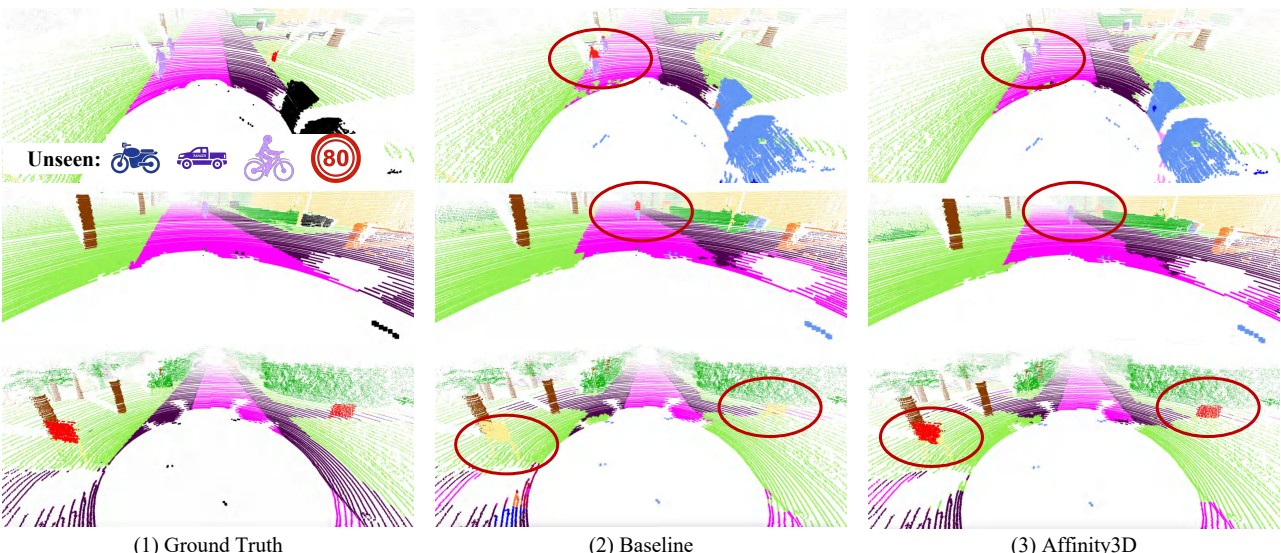

(1) Ground Truth
(2) Baseline
(3) Affinity3D

**Figure 5: The visualization results of our Affinity3D, baseline, and ground truth under a generalized zero-shot setting.**

that our affinity could be a quantitative indicator of object similarity. Fig. 5 compared our semantic segmentation results with the baseline. Affinity3D had fewer faults than the baseline. The baseline model in Fig. 5 incorrectly segmented most points belonging to the traffic sign and bicyclist class, whereas Affinity3D could maintain correct semantic segmentation results. In Fig. 5 rows 1 and 2, the baseline results struggle to maintain consistency at the instance level compared with Affinity3d. The baseline only correctly segments the top of bicyclists, which presents numerous fragmented predictions. The discerning ability of instance consistency validates the instance-level perception capability of Affinity3D. Accurate perception at the instance level benefited from our instance generation module, affinity module, and visibility measurement module.

## 5 CONCLUSION

In this work, we proposed a generalized zero-shot 3D semantic segmentation framework for progressively transferring knowledge from the image to the point clouds. The training framework improved 3D semantic segmentation performance without introducing additional parameters at inference time. Our framework generated more accurate pseudo labels at the instance level than previous methods. It leveraged the geometric spatial structure inherent in point clouds and the remarkable zero-shot properties of 2D vision-language models. The proposed affinity module enhanced the quality of pseudo labels by propagating similarity to other samples. The proposed visibility module measured the visibility of 3D points in camera view by comparing the depth of points with the corresponding superpixel's depth. It substantially softened semantic ambiguity by ignoring the invisible 3D points when transferring knowledge from images to point clouds. Our framework improved the SemanticKITTI dataset under generalized zero-shot and annotation-free settings. It got 63.65% hIoU under the generalized zero-shot setting and 18.48% mIoU under the annotation-free setting. Future research on transferring object composition knowledge from images to point clouds was still needed.

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
