# OpenReview forum: "Affinity3D: Propagating Instance-Level Semantic Affinity for Zero-Shot Point Cloud Semantic Segmentation"
_acmmm.org/ACMMM/2024/Conference — MM2024 Poster_

### Official Review · Reviewer_9CVq · 2024-04-28

**Rating:** 3
**Confidence:** 4

**Summary:**

This paper presents a framework called Affinity3D, which aims to enhance the ability of 3D semantic segmentation models to perceive new samples. The excellent geometric separation capability of 3D models and the zero-shot capability of 2D models are utilized to aggregate 3D instances into 2D models and recognize them. Affinity3D contains an affinity module that corrects erroneous predictions by comparing them with similar instances, and a visibility module that prevents knowledge from being transferred from visible 2D pixels to invisible 3D points. The proposed method is extensively experimented on the SemanticKITTI dataset and achieves state-of-the-art performance in both seen and unseen cases.

**Strengths:**

The novelty of this paper is debatable, as the techniques of CLIP, superpoint construction, 2D-3D projection, and transfer learning at the core of the paper are already in mature use. However, the affinity and visibility modules proposed in this paper seem to be only minor improvements to existing techniques or constraints on the data.

The theoretical approach and technicality of the paper is feasible, and the paper verifies this judgment of mine through sufficient theoretical knowledge and experimental results.

The graphical quality of the thesis is presented unevenly, with multiple fuzzy images, and I don't feel that the images need to be distorted. In addition, the depiction of the data formulas could be improved.

The applicability of the paper is noteworthy, as the zero-shot learning technique mentioned is a major challenge that needs to be overcome in the current field of 3D vision.

**Limitations:**

1. Affinity3D of this paper seems to be built on literature 5 and 26, however, they are partially ignored in the comparison results.

2. The first half of the introduction in the paper is too redundant, the authors should focus on the differences and connections between the previous methods or ideas and the proposed method, instead of just correcting the mistakes.

3. Why the input image of Fig. 1 is deliberately processed blurry, which seems to shed little light on the methodology of this paper, as the effect on 3D perception is not represented.

4. The text defines the data format of the point cloud as 𝑃 ∈ R𝑁 × 4, which should be stated as a combination of point coordinates and intensities, or just use 𝑃 ∈ R𝑁 × 3.

5. In section 3.7, the knowledge migration module seems to be in line with the idea of 2DPASS, both for networks and losses.

6. there are some grammatical errors or oddities in the paper, for example:
In the abstract, "from image to point clouds", "a visibility module", "SemanticKITTI datasets";
In the main text, "LiDAR and RGB images", "which pre-define classes", ", etc., are ", "3D points and pixels on 2D images", "segmentation heads", "a pseudo labels generation", "3D Backbone"’, "text editing function", "random walks", "'𝑠𝑐𝑎𝑡𝑡𝑒𝑟 𝑚𝑖𝑛'", "3D Points are projected to 2D pixel", "the image bound", "the font of equations 4 and 6".
More details need to be scrutinized again by the authors.

7. It is suggested to add related methods as comparison tests, such as “Hierarchical Point-based Active Learning for Semi-supervised Point Cloud Semantic Segmentation”, ICCV 2023; “Exploring Dual Representations in Large-Scale Point Clouds: a Simple Weakly Supervised Semantic Segmentation Framework", ACMMM 2023.

**Suitability:**

2

---

### Official Review · Reviewer_D6jy · 2024-05-18

**Rating:** 3
**Confidence:** 3

**Summary:**

This paper proposes Affinity 3D, which aims to make 3D semantic segmentation models aware of new samples. The framework aggregates instances in 3D and identifies them in 2D, taking advantage of the geometric separation in 3D and the zero-shot capabilities of 2D models. An affinity module is proposed to refine the error predictions of visual language models and generate accurate pseudo-labels. A visibility measurement module is proposed to avoid the misalignment of invisible three-dimensional points and achieve reliable knowledge transfer. Experiments prove that the proposed method achieves state-of-the-art results on the SemanticKITTI dataset.

**Strengths:**

1.There is a good motivation to transfer knowledge from CLIP to 3D to improve the zero-shot point cloud semantic segmentation task.
2.The issues addressed in the paper, wrong predictions and wrong geometric alignment, are insightful.

**Limitations:**

1.The method is only compared with the current state-of-the-art methods on the SemanticKITTI data set, which is not enough to prove the effectiveness of the method. Because comparisons on a single data set may have problems such as data set bias. It is recommended to refer to the method [1] and compare it with the current most advanced methods on data sets such as nuScenes.
2.The proposed method lacks complete and readable writing. For example, in section 3.4 distribution. The scale in Formula 5 and Formula 6 represents the affinity factor, whether this is a hyperparameter? Since this parameter is in the proposed Affinity module, there should be more explanation. And the experimental section should be supplemented with more experiments and parameter analysis of the scale.
3.Section 3.4 introduces: As illustrated in Fig. 4, if a particular instance yields a wrong prediction at time, corrections can be made by comparing instances across preceding and subsequent frames or considering contextual instances. However, Figure 4 is the visualize results of affinity. How does this explain the statement in the paper?
4.The figures in the paper are not enough to better explain the method, and some figures are poorly readable, causing confusion in understanding. Such as fig. 3 is quoted a total of three times in sections 1, 3.1 and 3.5 respectively, which causes a deviation in the understanding of the figure. If a figure has multiple meanings, please divide it into multiple subfigures, annotate each subfigure with a subtitle to explain the meaning of each part separately, and cite the subfigures in the text.
5.The methods section is not readable enough and the proposed methods are not clearly explained. Sections 3.2 and 3.3 only have one paragraph of introduction and are not enough to be listed as separate sections. If the method exists, it is recommended to merge it into Preliminary. Or adjust the section distribution as much as possible to increase the readability of the manuscript.

[1] Wang, Yuanbin, et al. "Transferring CLIP's Knowledge into Zero-Shot Point Cloud Semantic Segmentation." Proceedings of the 31st ACM International Conference on Multimedia. 2023.

**Suitability:**

2

---

### Official Review · Reviewer_zojY · 2024-05-24

**Rating:** 4
**Confidence:** 2

**Summary:**

The authors propose Affinity3D, a framework designed to enhance zero-shot point cloud semantic segmentation by effectively transferring knowledge from 2D models to 3D data. They introduce an affinity module to correct wrong predictions by comparing similar instances and a visibility module to prevent knowledge transfer from visible 2D pixels to invisible 3D points, addressing semantic ambiguity and visibility issues in the process. Extensive experiments are conducted on SemanticKITTI datasaet.

**Strengths:**

1. The proposed instance-based knowledge transfer has a clear motivation and is well-justified.
2. Filtering out invisible points in images to eliminate noise during the transfer process is novel and effective.
3. The method shows significant improvements over state-of-the-art methods for unseen categories on the SemanticKITTI dataset.

**Limitations:**

1. There are existing point-language pretraining models such as the PointCLIP series and ULIP. The authors did not compare their method with these approaches that do not require CLIP for knowledge transfer.
2. The paper only presents experimental results on the SemanticKITTI dataset and lacks a comparison with previous state-of-the-art methods on the nuScenes dataset.
3. The structure of the methods section is somewhat disorganized, comprising small modules that lack cohesion and connection, which negatively impacts the readability of the paper.
4. The different handling methods for seen and unseen categories are not clearly explained in the paper. The authors should provide a more detailed explanation of this aspect.

**Suitability:**

3

---

### Official Review · Reviewer_zZUH · 2024-05-24

**Rating:** 4
**Confidence:** 3

**Summary:**

This paper tackles zero-shot point cloud semantic segmentation task. Given that pseudo-labels generated by pretrained vision-language model are not accurate, an affinity module is proposed to refine wrong predictions based on random walk algorithm. A visibility measurement module is proposed to only align visible 3D points with 2D pixels, ensuring reliable knowledge transfer.

**Strengths:**

1. The motivation of this paper is clear and reasonable, and the proposed modules are effective.
2. The experimental results are good. This method outperforms previous methods under both GZS and AF settings.

**Limitations:**

1.	The experiments are only conducted on SemanticKITTI dataset. It is sugguested to add results on nuScenes dataset.
2.	The detailed explanation of Knowledge Transfer Module is hard to understand. The figure in the supplementary material doesn’t illustrate its detail, either. It is suggested to add more details about the module.

**Suitability:**

3

---

### Meta-Review · Area_Chair_pJSS · 2024-07-01

**Recommendation:** Accept (Poster)
**Confidence:** 5

**Metareview:**

This paper received three borderline accept and one borderline reject final ratings from the reviewers. The borderline reject reviewer main concern is that the quality of writing. But most reviewers can understand the content of this paper. AC agrees that this paper benefits from interesting idea. However, the authors are encouraged to make the necessary changes to the best of their ability.